# Integrated Crop-Livestock Systems for Nitrogen Management: A Multi-Scale Spatial Analysis

**DOI:** 10.3390/ani11010100

**Published:** 2021-01-06

**Authors:** Suraj Ghimire, Jingjing Wang, John R. Fleck

**Affiliations:** 1Department of Economics, University of New Mexico, Albuquerque, NM 87131, USA; fleckj@unm.edu; 2Water Resources Program, University of New Mexico, Albuquerque, NM 87131, USA

**Keywords:** integrated crop-livestock system, nutrient management, nitrogen, manure, spatial analysis

## Abstract

**Simple Summary:**

Manure disposal is a growing problem as agricultural specialization leads to ever-larger concentrations of farm animals. Animals and crops were once grown on the same farm, creating an easy path for manure disposal on cropland in a cycle from animals to feed crops and back. Increasing specialization today means that concentrated animal operations are no longer linked to adjacent cropland on which animal waste can be disposed, leading to significant off-farm externalities in the form of risks of air and water contamination. Using an arid lands case study of dairies and crop and grass land in New Mexico, USA, we explore the possibility of reintegration through the analysis of available crop and range land in the scale of counties and watersheds surrounding the state’s concentrated dairies. We find that there is often available land to make productive use of the waste. However, in developing the policy tools to reintegrate the animal waste-crop cycle among independent farms and ranches, it is critical to consider the appropriate geographic scale.

**Abstract:**

The size and productivity of the livestock operations have increased over the past several decades, serving the needs of the growing human population. This growth however has come at the expense of broken connection between croplands and livestock operations. As a result, there is a huge disconnect between the nutrient needs of croplands and the availability of nutrients from livestock operations, leading to a range of environmental and public health issues. This study develops a theoretical framework for multi-scale spatial analysis of integrated crop-livestock systems. Using New Mexico, USA as a case study, we quantify the amount of nitrogen produced by dairy farms in the state and examine if the available nitrogen can be assimilated by the croplands and grasslands across spatial scales. The farm-level assessment identifies that all the farms under study do not have adequate onsite croplands to assimilate the nitrogen produced therein. The successive assessments at county and watershed levels suggest that the among-farm integration across operations could be an effective mechanism to assimilate the excess nitrogen. Our study hints towards the multi-spatial characteristic of the problem that can be pivotal in designing successful policy instruments.

## 1. Introduction

The growing specialization of agriculture has broken a link that emerged over millennia of human farming—the connection between animals and crops. Today, we find a surplus of manure in animal operations, disconnected from the cropland where human and animal food is being grown. While the benefits in production of, and access to, food have been enormous, the resulting externalities in pollution and growing manure management costs highlight a coordination problem currently unresolved at either markets or governments. 

Croplands and livestock operations in the past were tightly knit around each other and formed a symbiotic system where the essential nutrients and organic matters for the crop growth were provided in the form of manure from livestock operations, and the forage and grains for the livestock were produced in the adjacent cropland [1]. A high degree of advancement in plant genetics, enabling technologies such as milking machines and synthetic chemicals, have drastically improved the agricultural yield, and have transformed the diverse agricultural farms of the past into specialized production facilities [2,3,4]. As farms consolidate and the less-productive operations are weeded out, both size and productivity of the farm grows [5]. This specialization has reduced food prices and increased the access to nutrition for a large swath of population but has also brought societal and environmental impairments such as air pollution from feedlots and lagoons, contamination of surface and groundwater, and problematic concentrations of manure in localized areas [3,4,6]. These industrial-scale animal operations are often established at strategic locations with cheap land prices, easy access to resources, lax regulations, and appropriate climate, which might not necessarily align with the acreage of croplands required to assimilate all the generated manure. The integration of an industrial-scale crop and livestock system is a behemoth task, and without competent management, can result in bottlenecks in the supply chain, suboptimal yields, and added burden to the farm’s bottom-line.

Furthermore, the specialized system of agriculture is a remnant of the time when energy and fertilizers were cheap, the climate was assumed to be static and the social and environmental cost of externalities were not a big concern [7]. However, the times have changed, energy is perceived to be costlier both in monetary terms and in terms of negative externalities associated with it, people are more aware of human-induced climate change, and there’s an even greater pressure from a growing human population and scarce water resources [7]. Therefore, what worked in the past is not likely to work in the future. Environmental policy interventions in response to contamination problems are generally disconnected from the agricultural policy interventions that inadvertently contributed to those problems, a classic policy coordination/integration problem [8,9]. Information on the central questions involved—Where is the manure? Where is the available crop and rangeland where it could be put to use?—is a necessary condition for building policy tools that bridge the divide.

Dryland regions occupy over forty percent of the land surface area and more than 2 billion people reside in these regions [10]. Up to 44% of the world’s cultivated area lie there and significantly contribute towards global food production [10]. The state of New Mexico in the United States (US), with its arid and semiarid climate and a significant intensification of dairy industry in the past few decades, stands out as a perfect candidate for the study of this trend. Although New Mexico ranks ninth in the US for the total number of milk cows, it has the largest herd size of 3187 cows and has held this position since 2002 [11,12]. The state had approximately 80,000 cows on 105 farms in the early 1990, which increased to 180 dairies with 325,000 dairy cows by 2007 and to 140 dairies with 326,000 dairy cows in 2019 [13]. Most of the large dairy farms in the state are concentrated in the state’s south and east—Chaves, Curry, Roosevelt, Doña Ana, and Lea counties—accommodating over 90% of the state’s cows [12]. The number of milk cows has recently increased by 6% from 2012 to 2017, but the number of dairy operations declined by 5%, which points toward a trend of dairy farm consolidation. Large dairy farms with 500 or more cows now account for more than 99% of the statewide inventory and occupy a comparable share of the dairy products market [12]. The dairy industry, including production, processing and export of the dairy products, is a major revenue source and employment generator for the state, but this growth comes at a cost of damages to public health and the environment [14]. 

Nitrogen (N) is one of the major nutrients found in dairy manure and wastewater. N enters a dairy farm as feeds and forages consisting of proteins and amino acids that contribute towards the maintenance, growth, and production of cattle [15]. A fraction of this N is converted into milk or meat, while a significant portion is excreted as urine or manure. Anthropogenic activities including excessive production and application of N feeds and fertilizers to increase the agricultural output have doubled the concentration of N in the environment over the past six decades [16]. While the increased agricultural output has availed to meet the ever-increasing global food demand, the residual N has become a leading pollutant of water, soil, and air. The livestock industry is one of the major contributors of N pollution. Nearly one-third of the global human-induced N emissions amounting to 65 Tg per year is contributed by this sector [17]. Dairy and beef cattle farms are responsible for nearly 44% of the total livestock-related global N emissions [17]. At small and manageable quantities, manure N boosts plant growth and productivity. However, the intensification of dairy farms has resulted in an oversupply of N in smaller land pockets, causing the degradation of environmental and human health. 

A study by Johnson et al. [18] indicates that nearly two-thirds of the New Mexican dairies had improper management of manure leading to groundwater pollution. This is concerning, because over 73% of the private and publicly supplied drinking water in New Mexico is sourced from groundwater [19]. Sandy and well-drained soils, as in the case of New Mexico, are at higher risk of seeping nutrients to underlying aquifers [20]. Major precipitation events can result in wastewater lagoon spills and wash away the applied nutrients from the croplands [21]. Elevated levels of nutrients such as N and phosphorus in surface water fuel the growth of harmful algal blooms and create hypoxic zones, which can negatively affect aquatic biodiversity. The toxins produced by these algal blooms can pose a serious risk to fish and other animals higher up the food chain. Exposure to or ingestion of these harmful algae or their microspores can severely impact human health, causing allergy, gastrointestinal illness, respiratory or neurological problems [22]. A high concentration of nitrate in the drinking water can cause blue-baby syndrome in infants [23,24,25]. Inconclusive evidence of negative birth outcomes and birth defects such as low birth weight, spontaneous abortion and infant mortality have also been associated with high nitrate levels in drinking water [26,27]. Furthermore, the economic damages of N pollution can include the cost of water treatment, loss of commercial fishing and shellfish production, loss of recreational revenues from activities such as fishing, boating etc. and the loss of real estate values [28].

A large number of researchers and entrepreneurs have been working to improve the environmental sustainability of agricultural systems by integrating crop and livestock operations [29,30]. The integration of crops and livestock provides opportunities for ecological interactions, increasing the efficiency of the system at nutrient recycling and enhancing the soil profile, thereby improving the net benefits for farmers [31]. The hot and dry regions of the southwestern US face some unique challenges towards the integration of crop-livestock systems. The economic and agronomic benefits of the integrated system in the arid and semi-arid region revolve mainly around the scarcity of water. The receding level of surface and underground water, the competing demand for water from oil industries, municipalities etc., the cost of installing and maintaining water delivery systems, transference of knowledge among stakeholders are some of the major challenges of integrated systems in the Southwest US [32]. 

Spiegel et al. [33] studied the possibility of integrating livestock operations with the croplands at multiple scales to meet environmental, production, and economic goals. The study extends from within farm integration to within county integration and looks at the possibility of intercounty integration. The study specifically identified the need for integration of livestock and farming operations in the southern plains of the US. Among-farm integration might be the only solution in areas where large concentrations of livestock strain the ability of any individual CAFO to secure adequate land locally for manure application [34]. Out of the total 730,000 hectares of croplands in New Mexico, 5% or 37,250 hectares were treated with manure in 2017 [12]. This was down from 49,000 hectares in 2002 [35]. Crop-livestock integration reduced water usage in the Texas high plains by 23% and nitrogen fertilizers use was reduced by about 40% compared to the monoculture cultivation [36]. Crops grown in sandy soil with integrated crop-livestock systems had similar yields as the comparable unintegrated systems with significant net benefits in terms of whole-system economic productivity, soil health and water conservation [37]. 

Stacey et al. [38] estimated N-balances between cropland use and manure production within Whatcom County, Washington. The N-balance equation suggested a small surplus of manure N in the county, ranging from 82 to 131% of cropland N needs. Porter and James [39] spatially modelled the application of manure nutrients to the proximal agricultural field in Minnesota. The study estimated the crop N requirements at varying application rates and quantified the assimilative capacity of croplands, taking manure haul distance as a constraint. Findings suggested that a few regions in the state had abnormally high manure production and a significant overapplication of N fertilizer was also occurring.

Lemaire et al. [40] suggested the need for local integration of crop and livestock systems to regulate biogeochemical cycles, increase biodiversity through diversified and structured landscape mosaic and increase the flexibility of the system to cope with potential hazards and crises. The study also proposed integrating grassland systems with the croplands to help mitigate negative environmental externalities. Utilizing the naturally growing grasses from the landscapes otherwise unsuitable for crop production diversifies the sources of forage for livestock and reduces the pressure on croplands. While the use of manure in croplands is not uncommon, proposals for applying manures to the arid grasslands usually devoid of soil amendments, is increasing [41]. The additional benefits of applying manure to rangelands include improved quantity and quality of forage for the livestock [41,42,43,44], reduced erosion of soil [41,45], increased activity and biomass of soil microbes and improved water retention [46].

Russelle et al. [29] studied the challenges and opportunities of reintegrating crop-livestock systems in North America. The study identified the economic and ecological benefits of crop-livestock integration and identified major hurdles in the widespread acceptance of manure. The cost of manure application increases as the willingness of neighboring crop producers to accept manure decreases. The cost of hauling increases and fuel price volatility adds another layer of uncertainty to the farm operation. Stenfield [47] argued that with time and sophistication, the crop-livestock integration will move from a local (within-farm) to a regional (among-farm) scale. The study concluded that despite numerous challenges in the integration of crop and livestock systems, the synergies created by the integration would provide a net benefit to all stakeholders.

Thornton [48] modeled the interactions of crop and livestock enterprises in the context of smallholder farming operations. The study identified minimum requirements and a generic modeling framework for the studies assessing the impact of crop-livestock integration. The effect of feed quality and storage methods on the quality and availability of manure nutrients also needs to be studied. The knowledge and perception of the farmers regarding these systems also affect their effectiveness. Kleijn et al. [49] identified anomalies in the perceived benefits of integrated production systems between scientists and farmers. More than the cost and benefits of these systems, farmers care about the ease of use and minimal interference to their day-to-day operations. In addition to the revenue, regulations and reputational concerns can also be equally important instruments towards a wide-scale adoption of integrated production systems. The study also identified a need for quantification of the economic benefits using farmer friendly variables such as crop yield and farm-level profits and that is relevant across a range of crops, farming systems and locations.

The farm-level study of a typical large dairy farm in New Mexico by Joshi and Wang, 2018 [11] suggested the need for a detailed study incorporating size range and spatial distribution of the farms. Our study fills in this gap by performing an N-balance assessment of individual farms with full information on herd and land sizes. Fleming et al. [50], Lazarus and Koehler [51], and Stacey et al. [38] recommend the judicious application of manure in cropland and rangeland to close the gap between agricultural nutrient demand and supply. One major contribution of this paper is that it identifies the possible integration of grasslands to form a crop-grassland-livestock system, by assessing the N-balance of the integrated system.

This study aims to assess the N-balance of integrated crop-livestock systems across multiple spatial scales. Specifically, we evaluate the capacity of croplands and grasslands to assimilate or recycle the available nitrogen in livestock manure. First, a theoretical framework is developed for multi-scale spatial analysis. Second, using the dairy sector in New Mexico as a case study, we employ the multiscale approach, starting at the farm-level focusing on within-farm integration. We then extend the analysis to the county and watershed-level to identify potential among-farm integration of crop-livestock systems. N-balance is assessed, and nitrogen hotspots are identified at each scale under the full integration of crop-livestock systems. Sensitivity analysis is performed to explore how the extent of integration impacts N-balance at the regional level.

We contribute to the literature and to ongoing policy discussions in at least three ways. First, the study presents a framework that can be applied to other regions interested in conducting similar assessments. Second, the study performs a multi-scale spatial analysis based on individual farm data. This enables a more accurate multi-scale analysis with less need for data approximation and extrapolation. Third, to our knowledge, this is the first study assessing the potential of integrated crop-grass-livestock systems for N management across multiple scales. The findings provide valuable insights to farmers as well as policymakers on using integrated crop-livestock systems to manage nutrients, especially to those in arid and semi-arid regions with both croplands and grasslands. 

## 2. Materials and Methods 

To assess the potential of integrated crop-livestock systems for nitrogen management, we employ a multi-scale spatial analysis of both within-farm and regional (among-farm) integration. The within-farm integration is assessed at the farm-level corresponding to boundaries of individual livestock farms. The regional integration is assessed at two alternative levels, one at the county-level corresponding to administrative boundaries, and the other at the watershed-level corresponding to hydrological boundaries. 

Nitrogen balance at each scale is defined as the difference between the total amount of nitrogen available for plant use from livestock manure and the total amount of nitrogen requirement for all crops cultivated. For a general setup, let h=1,…,H denote all types of livestock raised and g=1,…,G denote all types of crops (including forages and grasses) cultivated across all scales. For any livestock type h, $M_{h}$ denotes the average amount of manure produced and $N_{h}^{M}$ denotes the average amount of nitrogen available for plant use in the manure, which is estimated from the average nitrogen content of the manure and typical losses of nitrogen during manure storage, transportation, and application. For any crop type g, $N_{g}^{C}$ denotes the average amount of nitrogen required for crop growth. Each level of spatial analysis is developed below.

Nitrogen balance at the farm-level is assessed as in Equation (1), where $NB_{i}$ is the nitrogen-balance of farm i, $A_{i,h}$ is the population of livestock *h* raised on the farm, and $L_{i,g}$ is the onsite acreage of land for growing crop g. Given the general setup above, one should expect many zero values for $A_{i,h}$ and $L_{i,g}$ for a single farm.
(1)$$NB_{i}=\overset{H}{\underset{h=1}{\sum }}A_{i,h}M_{h}N_{h}^{M}-\overset{G}{\underset{g=1}{\sum }}L_{i,g}N_{g}^{C}$$

Nitrogen balance at the county-level is assessed as in Equation (2), where $NB_{j}$ is the nitrogen-balance of county j, $A_{j,h}$ is the total population of livestock h in the county, and $L_{j,g}$ is the total acreage of land for crop g in the county. Theoretically, county totals for livestock populations and crop acreages can be obtained by aggregating over all the farms in the county. In practice, when farm-level data is not available, county-level data on livestock populations and crop acreages can be used directly for nitrogen assessment at the county-level (while skipping the farm-level assessment). If both farm-level and county-level data are available, they can be used to cross-check the validity of the datasets.
(2)$$NB_{j}=\overset{H}{\underset{h=1}{\sum }}A_{j,h}M_{h}N_{h}^{M}-\overset{G}{\underset{g=1}{\sum }}L_{j,g}N_{g}^{C}$$

Nitrogen balance at the watershed-level is assessed as in Equation (3), where $NB_{k}$ is the nitrogen balance of watershed k,
$A_{k,h}$ is the total population of livestock h in the watershed, and $L_{k,g}$ is the total acreage of land for crop g in the watershed. Similar to the county-level analysis, watershed totals for livestock populations and crop acreages can be obtained by aggregating over all the farms in the watershed. In practice, when data are available at the county-level but not at the farm-level, watershed totals can be approximated by multiplying county totals by the percentage of the area of each county in the watershed.
(3)$$NB_{k}=\overset{H}{\underset{h=1}{\sum }}A_{k,h}M_{h}N_{h}^{M}-\overset{G}{\underset{g=1}{\sum }}L_{k,g}N_{g}^{C}$$

The multi-scale spatial analysis can be conducted at any time step, like seasonal or annual. In our case study, we perform the assessments on an annual basis based on the data characteristics of our studied region. Thus, all the variables and parameters are estimated as annual averages. We also conduct sensitivity analysis under varying levels of integration to explore how the extent of integration can affect the N-balance of integrated crop-livestock systems. 

This study uses dairy farms, croplands, and grasslands in New Mexico as a case study. Table 1 summarizes the descriptive statistics of our data. Data for individual dairy farms were obtained from the New Mexico Environment Department. The data are based on the records of wastewater discharge permits which are required by any dairy farms with significant wastewater discharge volumes and operate within the state. Available data include dairy farm address, operational status, the wastewater discharge volume, and onsite land application area in acres. The total number of cows per farm was obtained by dividing the permitted volume of wastewater discharge in gallons per day by the average gallons of wastewater generated per cow per day. The total number of cows per county and per watershed is obtained by aggregating the farm data based on their locations. Based on this information, New Mexico had a total of 310,375 cows across 139 dairy operations in 2019, which is very close to the 326,000 dairy cows officially reported [12], with the difference probably due to the inclusion of hobby farms in the official data. This validates our farm-level data. 

The onsite cropland acreage information is only available for 35 farms (out of the total 139), out of which 23 had nonzero acreages of onsite croplands and 12 had evaporation ponds only with no onsite croplands. All these farms are in dairy-concentrated counties, as shown in Figure 1. Therefore, any farm-level inferences drawn from them should be limited to dairy farms in those areas. 

The acreage data for different types of crops and grasslands were obtained from the land use land cover (LULC) data from the United States Department of Agriculture, which depict agricultural land cover at a 30 m resolution. The LULC data consist of a geo-referenced raster image with crop-specific categories and estimated acreages embedded. The data were analyzed using the software QGIS to calculate the acreages for each LULC category at different scales. The total area of New Mexico is more than 31.5 million hectares, out of which almost 2.1% or about 0.66 million hectares (fallow land included) was used as cropland in 2019. The selected single and double field crops in our study comprise more than two-thirds or about 0.45 million hectares of the croplands excluding fallow land. This number can vary year to year and is likely even greater as most of the fallow land is occupied by the field crops. Grasslands or pasture occupy almost ten times the acreages of field crops with a total of 4.65 million hectares. Shrublands constitute a major portion of the state, occupying about 64% of the land or 20.44 million hectares. The developed area which includes all the residential areas, industries, and other artificial structures comprises a tiny fraction of the land with about 386,880 hectares. Figure 2 displays the spatial pattern of croplands and grasslands in New Mexico.

To identify the acreage of cropland suitable for applying manure at the county and watershed level, we selected 12 distinct field crops and 5 double crops. Field crops such as corn, oats, wheat, hay, barley, and sorghum are recognized as significant manure recipient crops [4]. Fruit and vegetable crops do not generally receive manure due to the concerns of pathogen contamination. The field crops used in our N-balance assessment were winter wheat, alfalfa, sorghum, corn, hay, triticale, double-crop winter wheat/corn, double-crop winter wheat/sorghum, millet, oats, double-crop barley/corn, double-crop triticale/corn, barley, rye, spring wheat, durum wheat, and double-crop oats/corn in the descending order of their acreages. Acreage data for all three types of wheat (winter, spring, and durum wheat) were merged for calculating the acreage of wheat. The LULC data were also used to estimate the acreages of different crops on each dairy farm, as the dairy farm data from the New Mexico Environment Department do not include information on the types of crops grown in the onsite croplands. Calculating the farm-level acreages of each crop using the same approach as at the county or watershed level was prone to errors due to two main reasons. First, the spatial resolution of the LULC data is 30 m, which is comparable in dimension to the cropland acreages of dairy farms. Therefore, there is a high chance of misattribution and underestimation. Second, without identifying the actual plots of croplands, the only viable option is to draw a circle around the farm based on acreage data. However, the geographic coordinates of the farm are not necessarily in the geographic center of the farm. Therefore, spatial analysis using circles drawn around the farm might be unrepresentative of the onsite cropland. To circumvent these limitations, we used the proportions of crop acreages for three major crops at the watershed level to calculate the N utilization potential of the onsite croplands with non-zero acreage. We also performed an alternative assessment to see if using crops with the highest N requirement (e.g., what if corn is the only crop grown) would change the results.

Another piece of information required for a multi-scale spatial analysis is data on county and watershed boundaries. Shapefiles for the state and county-level administrative boundaries were downloaded from the website of the New Mexico Resource Geographic Information System. The surface water data were obtained from the United States Geological Survey (USGS). USGS hosts and maintains a national hydrological unit dataset known as the Watershed Boundary Dataset (WBD). The hydrologic boundaries in the WBD are determined based on topographic, hydrologic, and other relevant landscape characteristics without regard for administrative, political, or jurisdictional boundaries [52]. For this study, we used digitized 8-digit HUCs boundaries from the USGS. A total of 85 surface water sub-basins with corresponding 8-digit HUCs were identified in New Mexico. The groundwater data were obtained from the New Mexico Office of the State Engineer, which has delineated specific areas within the state with underlying groundwater sources as groundwater basins. The groundwater basin boundaries are independent on USGS hydrologic unit boundaries and may or may not overlap. A total of 39 groundwater basins were identified in New Mexico.

Figure 3 displays the surface water basins in New Mexico and Figure 4 displays the groundwater basins in New Mexico. Based on the observed dairy clusters along with a preliminary county-level assessment, six watersheds were recognized to accommodate the identified clusters and thus of interest to this study, among which four were surface water basins (Rio Grande-Albuquerque, Upper Pecos-Long Arroyo, Monument-Seminole Draws, and El Paso-Las Cruces) and two were groundwater basins (Curry and Portales). We chose to use groundwater basin boundaries instead of surface water basins for assessing the N balance of the regions surrounding Curry and Roosevelt Counties for two reasons. First, these areas are significantly away from major river systems and are heavily dependent upon the ground aquifer (the Ogallala aquifer) for consumptive and irrigation uses. Second, there are growing concerns of depletion and contamination of groundwater threatening the agriculture sector in the region due to a booming oil and gas industry.

The descriptive statistics of the studied watersheds are summarized in Table 1. Among the four surface water basins, the Rio Grande-Albuquerque basin is the largest, with a catchment area of 0.85 million hectares. However, with a total of 4646 cows, this basin also has the least spatial concentration of dairy farms. The Upper Pecos-Long Arroyo basin is the second largest in terms of area with 0.81 million hectares and has a cow population of 99,330. The El Paso-Las Cruces basin has a total of 10,301 cows and an area of 0.61 million hectares. The Monument-Seminole Draws basin, the smallest among the four, has 18,170 cows and occupies 316 thousand hectares. For the two groundwater basins, the Curry basin is the bigger one, occupying nearly 0.4 million hectares and accommodates 88,633 cows. The Portales basin has a total area of 162 thousand hectares and a cow population of 67,597.

The last piece of information required for our analysis is data on the amount of nitrogen available for plant use from dairy manure and nitrogen requirements for different types of crops. Parametric values related to the dairy manure characteristics were obtained from Todd et al. [53], which examined ammonia emissions and nitrogen partitioning of a typical dairy farm in New Mexico. The study quantified average daily excreted manure per dairy cow, N content of the manure and the percentage of N retained in manure and wastewater in the lagoons. Specifically, a cow is provided 0.6 kg of N as feed daily. Out of this, 43% is lost as ammonia, 36% is partitioned to manure/lagoons, 19% is converted to milk and 2% retained in the cow. Based on this, an average cow generates 0.22 kg per day or 78.84 kg per year of biologically available N for field crops. Parameters of crop nitrogen requirements follow Laboski et al. [54], where parameters are proposed for field, vegetable, and fruit crops with various soil types and growing conditions. For this, soil with organic matter below 2% and irrigation-based cultivation was selected, which meets the conditions of New Mexico. Table 2 summarizes the nitrogen requirements of different crops used in our analysis. Due to common nitrogen loss during and after land application (e.g., volatilization and leaching), we follow the recommendation of the New Mexico Environment Department to multiply the crop nitrogen requirement by 125% to obtain the N utilization potential of each crop.

## 3. Results and Discussion

The assessment of N-balance at the farm-level shows that each dairy farm under study produced more N than can be assimilated by the onsite croplands. In other words, there is a weak within-farm integration of the crop-livestock system in New Mexico. In the absence of within-farm integration, the role of among-farm integration becomes more pronounced. The assessment of N-balance at the county and watershed level shows that among-farm integration along with the inclusion of grasslands for manure application can be the viable solution to meet the N requirements of crops and sustainably manage the effluents from dairy farms. 

### 3.1. Farm-Level Assessment 

All the 23 dairy farms with non-zero acreages of onsite croplands showed positive N-balance when the average cropping pattern of their respective watersheds was used (see Table A1 in Appendix A). To check the robustness of this assessment, we assumed the crop with the highest N utilization potential (corn) to be grown in 100% of the cropland. This showed that all except one farm still had significantly positive N-balance (see Table A1 in the Appendix A). Similarly, all the 12 farms with zero acreages of croplands had positive N-balance by default. Positive N-balance indicates that a farm does not have enough onsite croplands to adequately assimilate the N available from manure. 

The acres of cropland for farms with nonzero acreage range from 36.83 to 382.83 hectares with a mean of 139. The number of cows in those farms range from 300 to 2700 with a mean of 1264 cows. Figure 5 presents a linear regression with the number of cows on a farm as an independent variable and the total acreage as a dependent variable. It shows that the crop acreage of a (non-zero acreage) dairy farm depends on the number of cows on that farm. For every additional cow, the average cropland size increases by 0.067 hectares. Moreover, even when we assume the crop with the highest N utilization potential (corn) to be grown in the additional 0.067 hectares for each cow, the N surplus increases by 64 kg/yr. This demonstrates a trend of intensification, as the increase in cropland size is not sufficient to assimilate the increased N loading from the increase in the number of cows.

With an average herd size of 1215, the average herd size of farms with zero acreages was similar to the farms with nonzero acreages. With 3200 and 4600 total cows, two farms in this subgroup were significantly larger than the largest farm in the nonzero acreage subgroup. Similarly, 3 farms had less than 300 cows in the zero-acreage subgroup. 

The average annual N available from the dairies with non-zero acreage was 99,685 kg, compared to 25,745 kg of N utilization potential. On average, dairy farms generated almost four times more N than what can be assimilated by the onsite croplands. However, it should be noted that some of the farms under study did not have any land application area and relied solely on the evaporation ponds to dispose of N, which is legal in New Mexico. The prospect of among-farm integration is even higher for these farms as they depend on other farms to dispose-off the manure solids generated within the farm.

Most of the management decisions, such as production levels, production costs, productivity, and profitability, are made at the farm-level. However, the environmental externalities of a farm extend to the territorial and regional levels, depending on the topographical, hydrological, and ecological systems [40]. Therefore, it is imperative to expand the analysis at the regional or territorial level to get a full picture of the threats and possibilities of crop-livestock integrations. 

### 3.2. County-Level Assessment 

Results from the farm-level assessment might overstate the incidence of positive N-balance due to the assumption that farms do not export manure to neighboring croplands. In the county-level assessment, we assume that all acres of the 12 distinct field crops and 5 double-crop combinations within the county boundary are available for manure application. Out of the 33 counties, 12 counties in New Mexico were identified as the dairy-concentrated counties based on the discharge permit issuance records. The number of cows in these counties ranged from 300 in Bernalillo county to 102,210 cows in Chaves County. Although the literature regarding the practice of among-farm integration in these regions is limited, most of the dairy-concentrated counties are also some of the biggest crop farming regions in the state. Chaves County was a notable exception, as the acreage of crops in the county was not sufficient to utilize the N generated from dairy farms. Doña Ana, although a major producer of fresh produce such as onion, pepper, and other crops such as cotton and pecan, had comparatively low acreages of the selected field crops in our study. 

Figure 6 displays the results of the N-balance assessment at the county level (see Table A2 in the Appendix A for detailed N balances). Chaves and Doña Ana are the only two counties with an N surplus: Chaves had a net positive N-balance of over 4 million kilograms, whereas Doña Ana had a surplus of nearly 400,000 kg. Sierra and Socorro Counties have marginally negative N-balances of below 150 thousand kilograms. Despite being two of the largest milk-producing counties in the state, both Curry County and Roosevelt County had the largest negative N-balances of above 10 million kilograms, due to large acreages of field crops in each county. Similarly, Quay County with a negative balance of over 8 million kilograms of N can be a big player in the intercounty among-farm integration of crop and livestock production systems, in the presence of a feasible manure hauling regime.

These results should be interpreted with caution, as we assume that all croplands within the county are available to accept manure. Sensitivity analysis is performed later to address the concerns related to varying levels of willingness to accept manure (WTAM) at croplands and grasslands. Moreover, these administrative boundaries do not necessarily align with the topography, catchment area, or the terrain of the landscape. Most of the dairy farms in the hotspot regions are along the county borderlines, such as between Curry and Roosevelt, Chaves and Eddy, and Doña Ana and Sierra, with some farms even sharing their land across the border. This makes county-level delineation and the subsequent N assessment unrealistic and a cause of impediment for the seamless among-farm integration. Therefore, in the next section, we perform a watershed-level analysis that provides a better avenue for among-farm integrations. 

### 3.3. Watershed-Level Assessment 

The watershed-level assessment was conducted in two stages for each watershed: (1) compare N available from dairies with N utilization potential of croplands in the watershed, and (2) compare N available from dairies with N utilization potential of grasslands in the watershed. Both stages assume full WTAM, which will be relaxed in the sensitivity analysis.

Figure 7 displays the result from the first stage of the watershed-level assessment, focusing on the assimilative capacity of croplands. Compared to the county-level assessment, the Upper Pecos-Long Arroyo region remains a hotspot due to its huge dairy population and comparatively low acreage of field crops. The region surrounding Doña Ana County represented by the El Paso-Las Cruces basin no longer has an N surplus. This is mainly due to the inclusion of croplands along the northern part of the basin outside of the Doña Ana County, which increased the croplands’ potential to assimilate available N. 

To assess the extent of intensification in the watersheds, we also calculated the assimilative capacity to available N (AC/AN) ratio, which is a ratio of the N utilization capacity of the croplands or grasslands and the available N from manure. Table 3 summarizes the results. Despite being one of the least productive regions in terms of net cash farm income and with the least acreage of croplands among the six basins, the Rio Grande-Albuquerque basin irrigated by the Rio Grande river is among the least vulnerable, with an AC/AN ratio of 2.44. The manure generated by a small cow population dispersed throughout the region can be reasonably assimilated into the existing cropland for the successful integration of the crop-livestock system in the region. The Curry basin has the largest AC/AN ratio of 3.14. In contrast to RGA, the Curry basin is one of the most productive agricultural regions, with almost 15 times more acreages of the selected crops. The major crops of this watershed include wheat, sorghum, and corn, which have a greater N utilization potential than alfalfa that is more prevalent in other regions. Therefore, despite having the highest concentration of cows, the Curry watershed has one of the lowest concerns of N surplus. This suggests that with a well-functioning among-farm integration mechanism, a region can be self-sufficient in assimilating the available N within the region. The remaining two watersheds, Portales and Monument-Seminole Draw, have AC/AN ratios of 1.35 and 1.49, respectively. Although sufficient to assimilate the available N, these margins are so low that they can be offset by small changes in the system, such as slight changes in the cropping pattern. 

In the second stage of the watershed-level assessment, we compare N available from dairies with N utilization potential of grasslands in each watershed, as summarized in Table 3. Results show that the grasslands in all the watersheds except the Portales basin could single-handedly assimilate the available N from manure. The Upper Pecos-Long Arroyo region, which is identified as the major hotspot of N surplus from all the previous analyses, has an AC/AN ratio of 3.14 after the inclusion of grasslands. When we add croplands into this mix, its AC/AN ratio increases to 3.56, providing sufficient room for the growing dairy industry in the region. The grasslands in the Rio Grande-Albuquerque and El Paso-Las Cruces basins have an AC/AN ratio greater than 26, indicating that these regions can accommodate the available N even at a dismal rate of WTAM. Grasslands in the Portales basin have an AC/AN of 0.70, which increases to 2.05 after including croplands, which might still be not enough to assimilate the available N at low WTAM levels. Inter-basin transfer of manure to the crop and grassland in Curry basin enables a combined AC/AN ratio of 3.64, suggesting potential successful management of manure in the region with a larger-scale integration of croplands, grasslands, and dairy farms. 

### 3.4. Sensitivity Analysis 

We conduct sensitivity analysis under varying levels of integration (i.e., alternative combinations of cropland and rangeland availability for manure disposal), to explore how the extent of integration can affect the N-balance of integrated crop-livestock systems. The analysis also provides a better understanding of the complexity of the integrated system.

The amount of land available for manure application can be limited by a range of factors, including the amount of land available, land cover, topography, depth of water table, locations of water bodies, local regulations, transportation costs, and crop producer’s preferences [34]. The preference of a crop producer might be shaped by a variety of factors, including the perceived risk of pathogens to farm staffs, neighbors, farm animals, and crops [55]; imperfect substitutability for commercial fertilizer, undesirable odor, weeds, soil compaction, soil pH alteration, inconsistent nutrient release, unavailability of specialized equipment, concerns of greater regulations, and environmental impacts [56]. 

Several studies identified the need for high WTAM values for the successful integration of crop-livestock systems, but advised against relying on fixed WTAM values [34,56,57]. To account for the uncertainties introduced by both farm and farmer level heterogeneities, we used 0%, 20%, 40%, 60%, 80%, and 100% levels of WTAM and reassessed the N-balance of the watersheds. A matrix was built with WTAM of grasslands on the *X*-axis and WTAM of croplands on the *Y*-axis and with ‘+’ or ‘−’ signs showing positive or negative N-balances. We created a unique vulnerability score to assess the capacity of each watershed in assimilating all the available N at varying WTAM levels. The score is based on the number of positive cells for a watershed out of the total possible score of 36. The vulnerability score for a region without any cow will be zero, and it will be 36 for a region that cannot assimilate N, even with its all croplands and grasslands. In sum, the higher the vulnerability score, the more vulnerable a region is to N surplus. 

Table 4 shows the N-balance of all six watersheds at varying levels of WTAM. With a score of 3, the Rio Grande-Albuquerque basin is the least vulnerable of the six watersheds. The Portales basin is the most vulnerable, with a score of 18. Although the Portales watershed has a negative N-balance at 100% WTAM of croplands, the assimilative capacity of the basin declines rapidly for lower WTAM values and is exacerbated by the low acreage of grasslands in the region. The Upper Pecos-Long Arroyo basin is the second most vulnerable watershed with a score of 11, however, huge potential for N assimilation conditional upon the integration of grasslands with the crop-livestock system provides a glimpse of silver-lining for the watershed. The Curry, Seminole-monument Draws, and El Paso-Las Cruces watersheds have moderate levels of vulnerability with the respective scores of 5, 4, and 4. 

## 4. Conclusions

At its root, the dairy manure problem as we now confront it is an economic coordination problem. There are benefits to be had through coordination, but we lack the institutional mechanisms needed to overcome the costs of carrying out the coordination [58]. Clearly, more successful integration of crop and livestock systems could be good for farm profits, the environment, and public health, but how do we move from the present suboptimal equilibrium to a better one? Here, consideration of scale becomes important. The scale of a single farm was sufficient through most of human history, but the recent agricultural trend of consolidation and specialization has introduced the requirement for regulatory intervention and collective action.

Our analysis suggests the multi-scale spatial dimensions of the manure problem—manure concentrated at relatively small spatial scales, while available crop and grass land for manure application is distributed at larger scales. Solving for the challenges posed by such cross-scale linkages has been identified as a central feature of successful resource management regimes [59]. In particular, traditional spatial governance boundaries—such as, in our case, United States counties—have been identified as insufficient as compared to watersheds, a more natural spatial framework for analyzing and responding to resource and environmental challenges [60].

Environmental policies in relation to manure contamination challenges reflect the traditional menu of policy options: direction regulation (command-and-control), provision of information to assist coordination, tax, and financial incentive policies, and trading instruments [11,61]. In the United States, agricultural policies impacting the concentration of dairy production generally happen at the national level, primarily through inclusion in the Farm Bill, which is renewed every five years and was most recently completed in 2018 [62]. An agriculturally related environmental policy is also implemented at the national level, as through the Clean Water Act, which imposes practices and standards on concentrated animal feeding operations, and the Environmental Quality Incentives Program, which provides financial incentives to encourage better manure management practices [63]. Recognizing the value of institutional diversity [59], we make no recommendation about any preferred policy mechanisms. Instead, our analysis suggests important multi-scale spatial characteristics of the problem that need to be taken into consideration in the design of successful policy instruments. Our proposed framework and multi-scale approach can be adopted by regions interested in exploring such spatial characteristics. 

Analysis done at the watershed scale, rather than at the more traditional county scale often used in the collection and analysis of agricultural statistics, suggests the importance of a policy design that considers the availability of cropland and grassland on the broadest possible spatial scale. First, it allows identification of hot spots—places where the concentration of dairies or other manure-producing concentrated animal agriculture produces an excess of manure relative to on-farm assimilative cropland capacity. In our above case study of New Mexico, for example, analysis suggesting an excess of manure production in Doña Ana County relative to available cropland changes when neighboring lands in different counties, but the same watershed show nitrogen deficits that could be filled by an integrated system across sufficiently large spatial scales. Furthermore, identifying the areas of manure concentration and available cropland and grassland at various adjoining spatial scales can help policymakers choose among a range of policy options beyond the land application, including integrated anaerobic digesters associated with bioenergy production.

As resource managers and policymakers develop and maintain better geographical information systems documenting resources (including recycled resources such as nutrients in manure and produced water from fracking) characteristics at various scales, a potentially fruitful line of research would investigate the optimal portfolio of policy instruments across different scales. Open questions for future research include determining the factors impacting farmers’ willingness to accept manure, neighborhood effects, and the impact of the information provision approach on coordination. 

## Figures and Tables

**Figure 1 animals-11-00100-f001:**
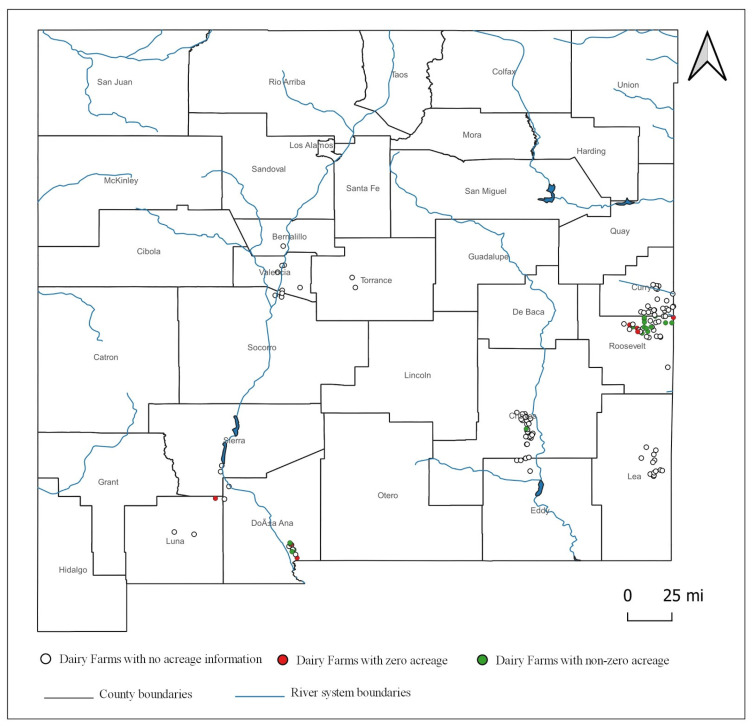
Dairy farms in New Mexico, US.

**Figure 2 animals-11-00100-f002:**
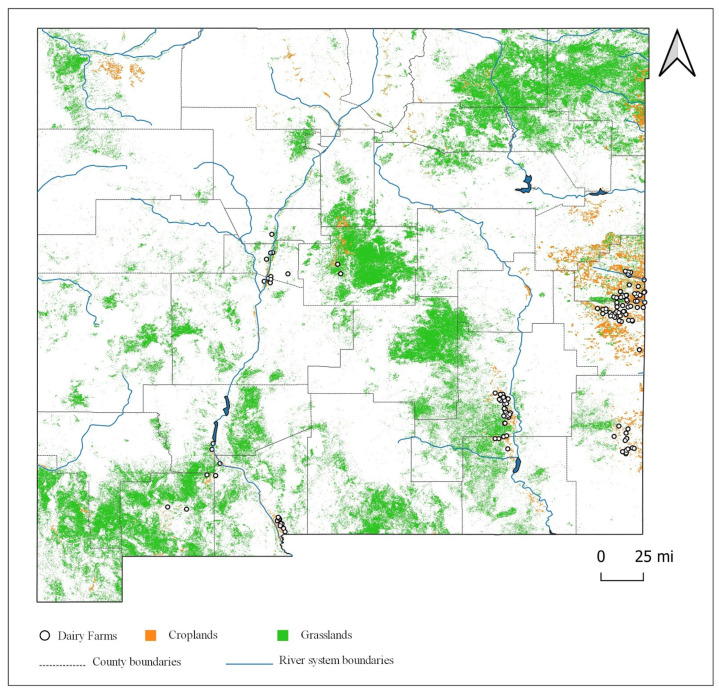
Croplands and grasslands in New Mexico, US.

**Figure 3 animals-11-00100-f003:**
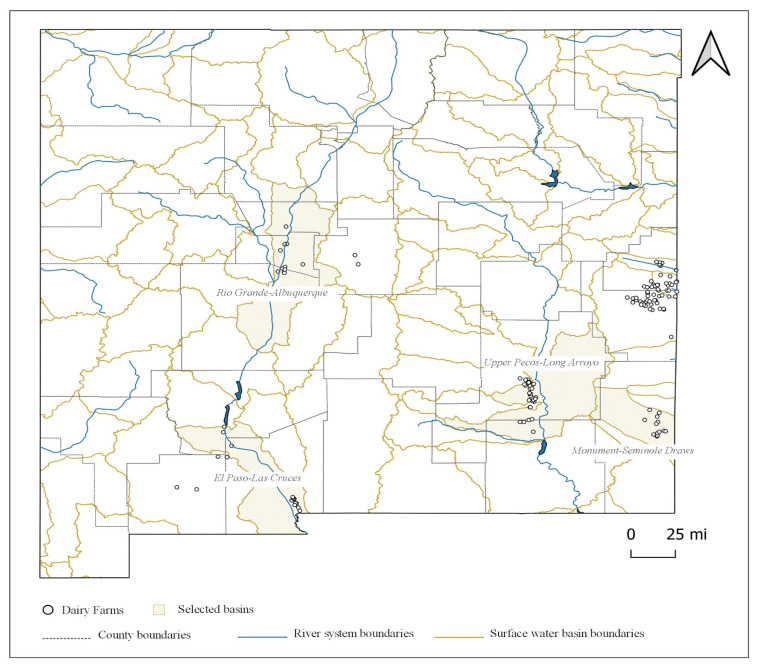
Surface water basins in New Mexico.

**Figure 4 animals-11-00100-f004:**
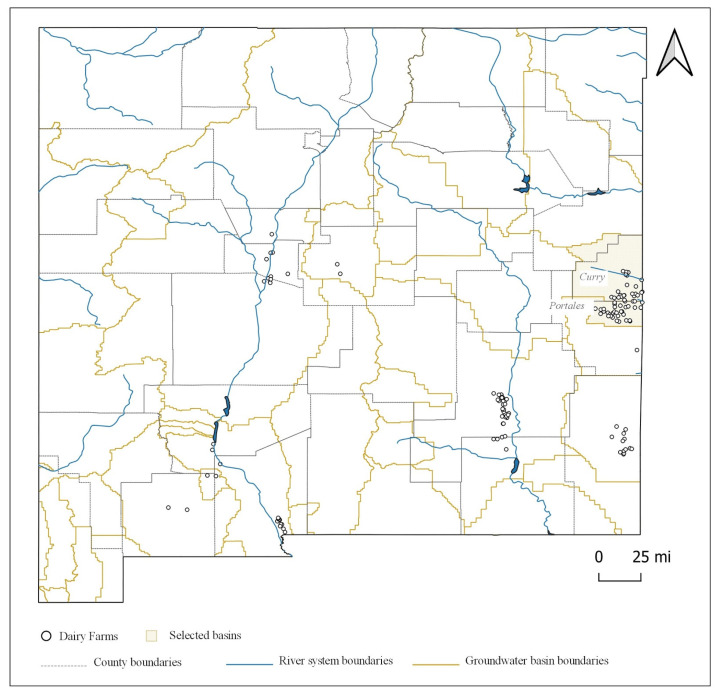
Groundwater basins in New Mexico.

**Figure 5 animals-11-00100-f005:**
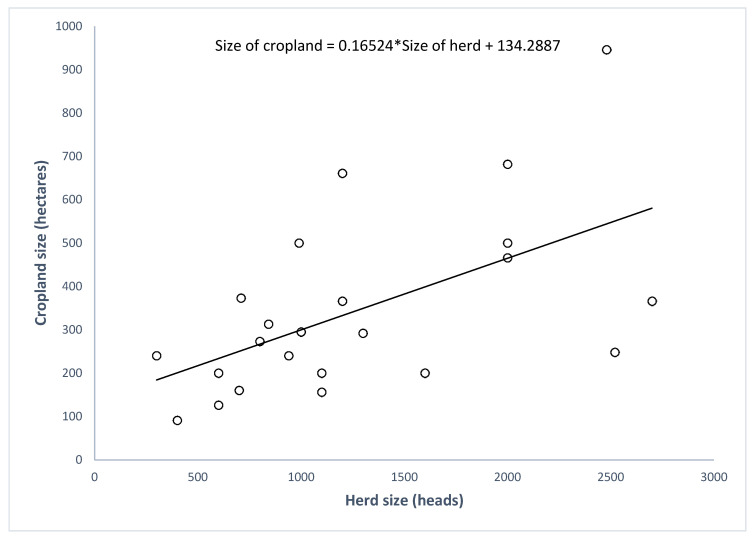
Correlation between herd size and cropland size.

**Figure 6 animals-11-00100-f006:**
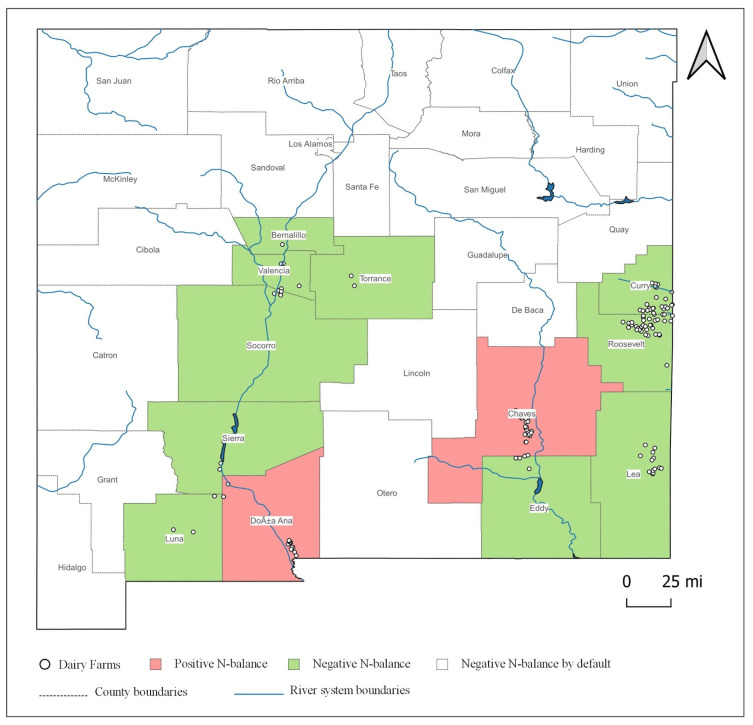
Nitrogen balance assessment at the county level.

**Figure 7 animals-11-00100-f007:**
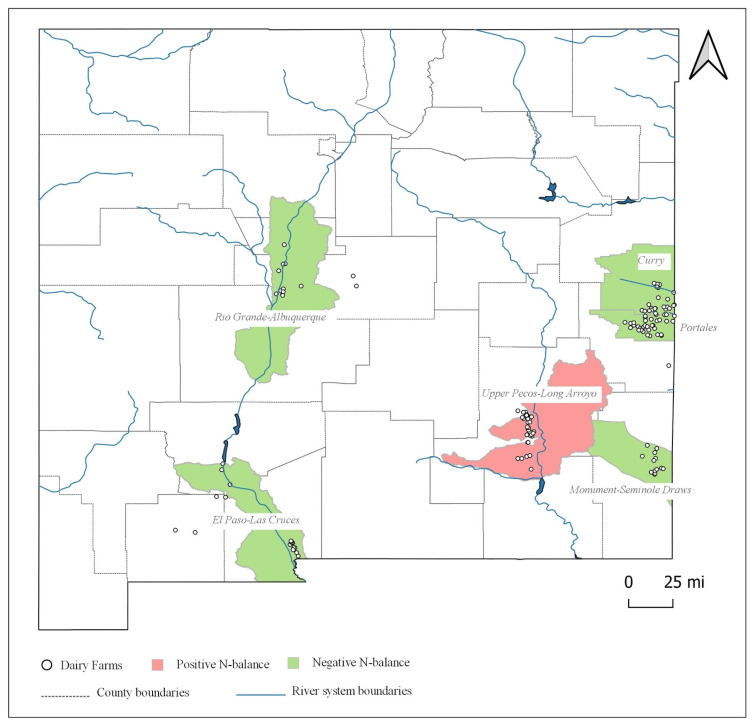
Nitrogen balance assessment at the watershed level.

**Table 1 animals-11-00100-t001:** Summary statistics of dairy farms, croplands, and grasslands in New Mexico.

	Units	Minimum	Maximum	Average	Total
Farm-level					
Cows	heads	40	40,000	2235	310,735
Cropland area ^1^	hectares	37	383	139	3195
County-level					
Cows ^2^	heads	300	102,210	9418	310,735
Cropland area ^2^	hectares	403	102,965	22,995	275,944
Grassland area ^2^	hectares	13,659	382,738	148,580	1,782,954
Watershed-level					
Cows ^3^	heads	4646	99,330	48,113	288,677
Cropland area ^3^	hectares	8520	124,166	34,953	210,590
Grassland area ^3^	hectares	16,596	109,742	61,227	367,364

Notes: ^1^ Exclusively for the 23 dairy farms with non-zero acreage of croplands. ^2^ The number of cows and acreage information is exclusively for the 12 dairy-concentrated counties. ^3^ The number of cows and acreage information is exclusively for the selected six watershed regions.

**Table 2 animals-11-00100-t002:** Crop nitrogen requirements.

Crop	Expected Yield Range (ton/hectare)	N (kg/hectare)
Alfalfa ^1^	2.47–6.18	16.80
Corn	13.56	242.16
Wheat	3.9	117.62
Grass (hay ^2^ and pasture grass ^3^)	1.24–19.77	179.15
Millet	2.24–3.36	89.7
Barley	1.34–5.38	79.07
Oat	1.14–4.57	67.21
Rye	0.94–4.39	67.21
Sorghum	3.14–6.28	145.71
Triticale	1.12–5.60	67.21

Notes: ^1^ Mature alfalfa does not require an N supplement, but the seedling requires 13.6 kg/acre. We take 6.8, the average of 0, and 13.6, assuming the mature and seedling alfalfa are equally distributed. ^2, 3^ Parameters related to Fescue grass are taken, which is the most widely grown hay crop in New Mexico. Pasture grasses including bromegrass, fescue, orchard grass, ryegrass, and timothy also have the same parameter value.

**Table 3 animals-11-00100-t003:** Watershed-level nitrogen balance assessment (in thousand kgs).

	AN ^1^	Crop AC ^2^	Grass AC	Crop NB ^3^	Grass NB	Crop AC/AN Ratio	Grass AC/AN Ratio
Surface Water basins							
RIO GRANDE-ALBUQUERQUE	366	894	9679	(527)	(9313)	2.44	26.42
UPPER PECOS-LONG ARROYO	7831	3277	24,576	4484	(16,744)	0.42	3.14
MONUMENT-SEMINOLE DRAWS	1433	2137	8939	(704)	(7507)	1.49	6.24
EL PASO-LAS CRUCES	812	1110	23,410	(297)	(22,597)	1.37	28.82
Groundwater Basins							
CURRY	6988	21,963	11,948	(14,975)	(4960)	3.14	1.71
PORTALES	5329	7174	3716	(1846)	1613	1.35	0.70

Notes: ^1^ AN = Available Nitrogen ^2^ AC = Assimilative Capacity ^3^ NB = Nitrogen Balance.

**Table 4 animals-11-00100-t004:** Watershed-level nitrogen balance under alternative WTAM of croplands and grasslands.

	Grasslands							
**Croplands**	**Watersheds**	**WTAM %**	**0**	**20**	**40**	**60**	**80**	**100**
	Rio Grande-Albuquerque	0	+	-	-	-	-	-
		20	+	-	-	-	-	-
		40	+	-	-	-	-	-
		60	-	-	-	-	-	-
		80	-	-	-	-	-	-
		100	-	-	-	-	-	-
	Upper Pecos-Long Arroyo	0	+	+	-	-	-	-
		20	+	+	-	-	-	-
		40	+	+	-	-	-	-
		60	+	+	-	-	-	-
		80	+	+	-	-	-	-
		100	+	-	-	-	-	-
	Curry	0	+	+	+	-	-	-
		20	+	+	-	-	-	-
		40	-	-	-	-	-	-
		60	-	-	-	-	-	-
		80	-	-	-	-	-	-
		100	-	-	-	-	-	-
	El Paso-Las Cruces	0	+	-	-	-	-	-
		20	+	-	-	-	-	-
		40	+	-	-	-	-	-
		60	+	-	-	-	-	-
		80	-	-	-	-	-	-
		100	-	-	-	-	-	-
	Monument-Seminole Draw	0	+	-	-	-	-	-
		20	+	-	-	-	-	-
		40	+	-	-	-	-	-
		60	+	-	-	-	-	-
		80	-	-	-	-	-	-
		100	-	-	-	-	-	-
	Portales	0	+	+	+	+	+	+
		20	+	+	+	+	+	+
		40	+	+	+	+	-	-
		60	+	+	-	-	-	-
		80	-	-	-	-	-	-
		100	-	-	-	-	-	-

## Data Availability

Data is contained within the article.

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
