# Peer review of "Integrated Crop-Livestock Systems for Nitrogen Management: A Multi-Scale Spatial Analysis"

_animals, 2021, doi:10.3390/ani11010100_

Round 1

Reviewer 1 Report

I have read the article with great pleasure and I found it very interesting. I believe it is a study that can be of a high interest for the scientific community and for the policy makers, due to the useful policy implications. The article is well written, balanced and uses a robust and accurate methodology. It takes into account all the relevant aspects from an economic, environmental and organizational point of view. I do not have any specific suggestion to improve the article. I believe it can be published in its actual form.

Reviewer 2 Report

The authors must adopt the International Systems of Unit. Use hectares instead of acres.

Tabel diagramtion must be improved

Reviewer 3 Report

The paper deals with an interesting topic, i.e. the integration of crop and livestock systems in a wide area and the needing of the analysis through a multi-scale approach to individuate intervention for regulating the manure problem.
The paper is well written, clear and effective. Anyway, in the tables and figures often the units of measure do not appear, and the American and international metric systems are used. It would be necessary to be more precise and standardize.
